# Mitigating Non-Uniform Forgetting Dynamics for Class-incremental Semantic Segmentation

## Abstract

Class-Incremental Semantic Segmentation (CISS) aims to learn newly introduced classes sequentially while preserving performance on previously learned ones. Most existing methods mitigate catastrophic forgetting through pseudo labels or regularization, but largely assume that forgetting evolves uniformly across old classes. In this paper, we reveal and characterize a *Non-Uniform Forgetting* (NUF) phenomenon in CISS, where different old classes exhibit markedly different forgetting trajectories in terms of degradation severity, onset time, and temporal pattern. Our analysis further shows that NUF is closely related to semantic complexity, semantic overlap, and the inherent old–new supervision imbalance of CISS. To address this problem, we propose a pseudo-labels-assisted framework with two complementary components. The first, *Imbalance-Aware Gradient Defence* (IGD), alleviates optimization bias through pixel-wise gradient-aware reweighting and channel-wise balancing, while a background-suppression term further reduces spurious foreground activations. The second, *Representation Drift Suppressor* (RDS), improves representation stability by enhancing inter-class separability with prototype-based contrastive learning and preserving old semantics through selective decoder-level distillation. By jointly combining IGD and RDS, the proposed framework effectively mitigates heterogeneous forgetting and yields more balanced incremental segmentation performance. Extensive experiments on PASCAL VOC and ADE20K under multiple incremental protocols demonstrate that the proposed method consistently improves old-class retention and overall incremental performance, outperforming state-of-the-art CISS approaches.

## 1 Introduction

Semantic segmentation aims to assign a semantic class label to each pixel and serves as a fundamental component of fine-grained scene understanding in modern visual perception systems (Long et al., 2015; Ma et al., 2023; Wang et al., 2024). In practical applications, segmentation models are often required to continually acquire novel object classes over time, without retaining full access to the training data of previously learned classes. This requirement gives rise to Class-Incremental Semantic Segmentation (CISS), where learning proceeds sequentially: at each incremental step, pixel-level annotations are provided only for newly introduced classes, while pixels belonging to previously learned classes, future unseen classes, and true background are all annotated as background. Under this protocol, old classes lose direct supervision in subsequent steps and are therefore progressively overwritten by the optimization toward newly introduced classes, leading to catastrophic forgetting (McCloskey & Cohen, 1989).

To mitigate forgetting, most existing CISS methods rely on pseudo labels strategies (Douillard et al., 2021; Yuan et al., 2024; Zhao et al., 2023) or regularization-based constraints (Cermelli et al., 2020; Xiao et al., 2023; Shang et al., 2023). Although these approaches have achieved promising results, they largely share an implicit assumption that forgetting can be treated homogeneously across all old classes. In practice, they typically apply global regularization or uniform pseudo labels to old classes as a whole, without distinguishing their different vulnerability levels. As a result, the protection provided by such methods is inherently coarse-grained and may be mismatched to the actual degradation dynamics of CISS.

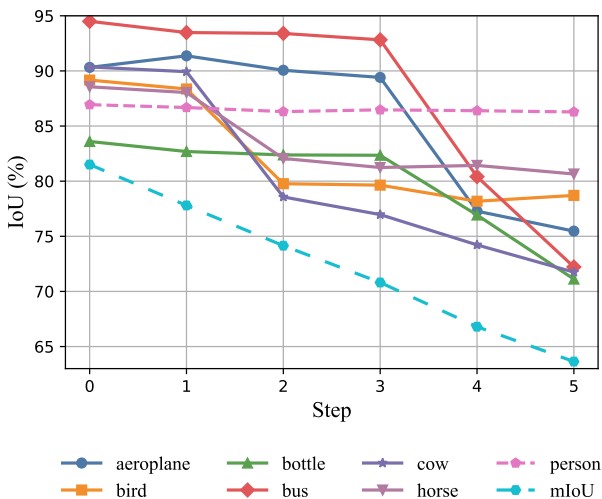

Figure 1: Illustration of non-uniform forgetting (NUF) in CISS under the VOC 15–1 protocol. Different old classes exhibit markedly different forgetting trajectories over incremental steps: while some classes remain relatively stable, vulnerable classes such as *cow* and *aeroplane* undergo early and severe degradation. This class-wise heterogeneity cannot be adequately reflected by aggregate metrics such as mIoU alone.

In this work, we show that forgetting in CISS is intrinsically heterogeneous rather than homogeneous. Our empirical analysis reveals that different old classes exhibit markedly different forgetting trajectories during incremental learning. As illustrated in Fig. 1, some classes, such as *person*, remain relatively stable across incremental steps, whereas others, such as *cow* and *aeroplane*, undergo rapid and substantial degradation shortly after losing supervision. We term this phenomenon *Non-Uniform Forgetting* (NUF). Specifically, NUF is manifested in class-dependent differences in forgetting severity, onset time, and degradation pattern. These observations suggest that the central limitation of existing CISS methods is not merely insufficient supervision, but their structural mismatch with the heterogeneous nature of forgetting. These also highlight the limitation of relying solely on aggregate metrics such as mIoU, which tend to smooth over critical class-wise dynamics and may conceal behaviors essential for understanding and mitigating catastrophic forgetting.

To address this problem, we propose a pseudo-labels-assisted CISS framework with two complementary modules. The first, **Imbalance-Aware Gradient Defence** (IGD), targets the optimization bias induced by severe old-new imbalance. It strengthens vulnerable old-class supervision through pixel-wise gradient-aware reweighting and channel-wise balancing, while a background-suppression term further reduces spurious activations of newly introduced classes. The second, **Representation Drift Suppressor** (RDS), targets the gradual drift of previously learned representations that cannot be fully resolved by pseudo-label correction alone. It improves cross-class separability through prototype-based contrastive learning and stabilizes old semantics through selective decoder-level distillation. By jointly integrating IGD and RDS, the proposed framework effectively alleviates catastrophic forgetting and its non-uniform behaviors, leading to more balanced and robust incremental segmentation performance.

Our main contributions are summarized as follows:

- We uncover and formalize the phenomenon of NUF in CISS by systematically analyzing class-wise degradation trajectories. To the best of our knowledge, this is the first study to explicitly characterize and quantify the heterogeneous forgetting dynamics of old classes in CISS.

- We propose a pseudo-labels-assisted framework to mitigate NUF through two complementary modules. IGD alleviates optimization bias through gradient-aware pixel reweighting, channel-wise balancing, and background suppression, while RDS suppresses representation drift through prototype separation and selective decoder-level distillation.

- Extensive experiments on two widely used benchmarks under multiple incremental protocols demonstrate that the proposed method consistently improves old-class retention and overall incremental balance, especially under more challenging long-horizon settings.

## 2 Related Work

### 2.1 Continual Learning

Continual Learning (CL), also known as incremental learning, aims to acquire new knowledge from streaming data while mitigating catastrophic forgetting. Existing CL research, primarily developed for image classification, can be broadly categorized into regularization-based, architectural, and replay-based approaches. Regularization-based methods Kirkpatrick et al. (2017); Kang et al. (2022); Gao et al. (2024; 2025) constrain parameter updates through explicit penalties or distillation losses. Architectural approaches Rusu et al. (2016); Ni et al. (2025) alleviate interference by expanding or reconfiguring network structures for task-specific adaptation. Replay-based methods Rebuffi et al. (2017); Aghasanli et al. (2025) reduce forgetting by storing exemplars or synthesizing pseudo-samples for joint training on previously learned and newly introduced classes.

Beyond these general paradigms, recent studies have begun to examine the role of class imbalance and heterogeneous forgetting in class-incremental learning. For example, He et al. He (2024) propose dynamic gradient reweighting to alleviate the optimization bias toward new classes. Yin et al. Xu et al. (2024) introduce prototype-based calibration to correct decision-boundary shifts caused by imbalance. Dong et al. Dong et al. (2023) design gradient-magnitude compensation to protect vulnerable classes. These studies suggest that forgetting is often class-dependent rather than uniform, and that effective mitigation should explicitly account for unequal vulnerability across classes.

Motivated by these observations, we investigate NUF in CISS. Compared with CL, CISS is more challenging because it lacks exemplar replay in the standard setting and relies heavily on pseudo supervision for old classes. As a result, class-specific degradation can be further amplified by pixel-level imbalance, background ambiguity, and noisy pseudo labels. Our work extends the study of heterogeneous forgetting from classification to dense prediction and develops mitigation strategies tailored to the unique challenges of CISS.

### 2.2 Class-Incremental Semantic Segmentation

CISS extends class-incremental learning from image-level recognition to dense prediction, requiring models to learn newly introduced classes while preserving fine-grained knowledge of previously learned ones. Early work such as MiB Cermelli et al. (2020) identified background drift as a central challenge and introduced unbiased cross-entropy together with unbiased distillation. Subsequent methods have approached forgetting mitigation from different perspectives. Pseudo-labels-based and distillation-based methods, such as PLOP Douillard et al. (2021), DKD Baek et al. (2022), SSUL Cha et al. (2021), and SVSRD Chang et al. (2025), improve old-class preservation by refining supervision transfer, background handling, or relational consistency across scales. Feature-regularization methods, such as SDR Michieli & Zanuttigh (2021), exploit contrastive learning to encourage more discriminative representations, while RCIL Zhang et al. (2022a) and EWF Xiao et al. (2023) further strengthen incremental adaptation through compensation or dynamic fusion mechanisms. Another line of research explores prototype replay or feature replay, as in STAR Chen et al. (2023) and its extensions Zhu et al. (2025); Zhu et al., which use stored prototypes or generated pseudo-features to assist incremental learning. In this work, we focus on the exemplar-free CISS setting.

Most existing CISS methods still assume approximately homogeneous forgetting across old classes and therefore adopt class-independent compensation. However, old classes often exhibit distinct degradation trajectories that cannot be adequately handled by uniform pseudo labels or step-level correction alone. This leads optimization to favor dominant classes while leaving semantically fragile classes under-protected. Our work instead explicitly addresses NUF in CISS by mitigating old–new imbalance in optimization and stabilizing old representations against cross-step drift.

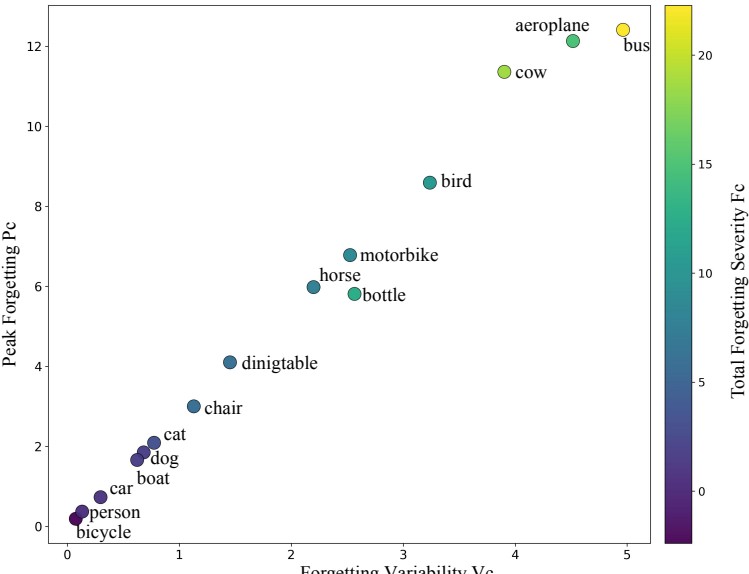

Figure 2: Quantitative characterization of NUF on VOC 15–1. Each point denotes a base class. The x-axis shows forgetting variability $V_c$ (the standard deviation of per-step IoU drops), the y-axis shows peak forgetting $P_c$ (the maximum one-step drop), and the color encodes total forgetting severity $F_c = IoU_c^{\text{init}} - IoU_c^{\text{final}}$.

## 3 Observation and Motivation

### 3.1 Quantifying Non-Uniform Forgetting

To systematically investigate forgetting in CISS, we begin with the simplest baseline setting, namely $L_{\text{mbce}} + L_{\text{KD}}$, and examine the class-wise IoU trajectories of all base classes under the VOC 15–1 protocol, as shown in Fig. 1. The results reveal that forgetting is highly non-uniform across classes in three aspects. First, the severity of degradation varies substantially: some classes suffer dramatic performance drops, whereas others remain largely stable throughout incremental learning. Second, the onset of forgetting is clearly asynchronous: certain classes collapse shortly after the first few increments, while others deteriorate only at much later stages. Third, the degradation pattern itself is heterogeneous: some classes exhibit gradual decline, whereas others remain stable for several steps and then undergo abrupt collapse. Moreover, once such a collapse occurs, the lost performance is rarely recovered in subsequent increments. These observations suggest that overall mIoU, as an averaged statistic, conceals substantial class-specific heterogeneity in both the magnitude and temporal structure of forgetting.

To quantify this phenomenon, we first define the total forgetting severity of class $c$ as $F_c = IoU_c^{\text{init}} - IoU_c^{\text{final}}$. As shown in Fig. 2, $F_c$ varies markedly across classes: vulnerable classes such as *cow* and *aeroplane* degrade by more than 20%, whereas stable classes such as *person* and *bicycle* drop by less than 5%. This nearly five-fold gap confirms that forgetting in CISS is strongly class-dependent rather than uniformly distributed.

However, cumulative forgetting alone cannot capture when and how degradation unfolds. We therefore define the per-step IoU drop as $d_{c,t} = \max\left(0, IoU_c^{(t-1)} - IoU_c^{(t)}\right)$, and introduce two complementary temporal indicators: Peak Collapse Severity, $P_c = \max_t d_{c,t}$, and Temporal Variability, $V_c = \text{Std}(\{d_{c,t}\}_{t=1}^T)$. Here, $P_c$ measures whether forgetting is concentrated in a single catastrophic step, whereas $V_c$ captures the unevenness of degradation over time. As illustrated in Fig. 2, classes with large $P_c$ usually also exhibit high $V_c$, indicating that temporal instability in CISS is mainly driven by localized collapse events rather than smooth cumulative

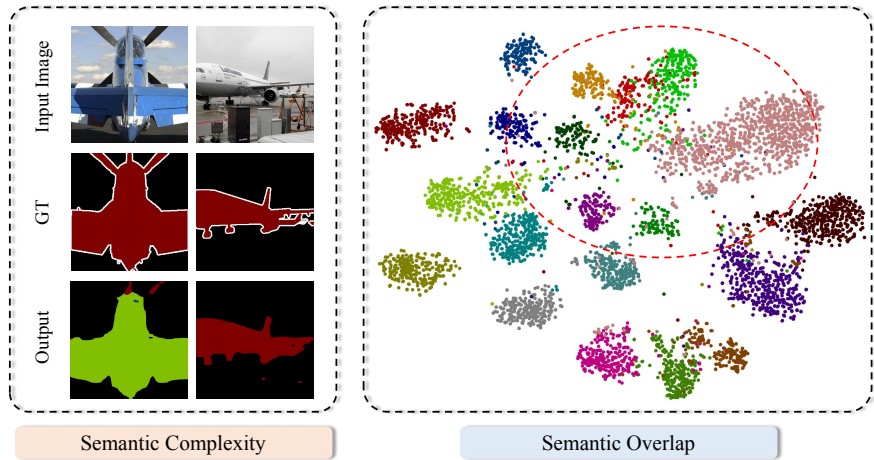

Figure 3: Two factors underlying NUF in CISS. Left: classes with high semantic complexity exhibit large intra-class variation, making their representations harder to maintain once dense supervision is removed. Right: semantic overlap causes old and new classes to reside in entangled feature regions, where optimization for new classes can interfere with old-class representations and trigger drift.

erosion. Together, $\{F_c, P_c, V_c\}$ show that NUF in CISS is manifested not only in how much each class forgets, but also in when forgetting occurs and whether it takes the form of gradual drift or abrupt collapse.

## 3.2 Analyzing the Causes of NUF

While the above results establish that forgetting in CISS is highly non-uniform, they do not explain why different classes follow such distinct degradation trajectories. To better understand the origin of this phenomenon, we further analyze the intrinsic factors that govern class-wise forgetting behavior. Importantly, these factors do not act in isolation. Under the CISS annotation protocol, newly introduced classes dominate each incremental step with abundant pixel-level supervision, whereas previously learned classes appear only sparsely and are supervised, at best, by pseudo labels. This severe pixel-level imbalance makes continual optimization inherently biased toward new classes, thereby amplifying the vulnerability of old classes with unfavorable semantic properties.

**(i) Semantic complexity.** Classes with high semantic complexity, typically reflected in large intra-class variability, are harder to encode into compact and stable representations. For instance, *aeroplane* exhibits substantial variations in viewpoint, geometry, and appearance. As shown in Fig. 3(left), even strong methods such as SVSRD struggle to maintain consistent predictions across different instances of this class. In contrast, structurally regular classes such as *person* tend to form more compact representations and thus remain more stable during incremental learning. Once pixel-level supervision is no longer available, semantically complex classes are more likely to undergo progressive drift or fragmentation, leading to early degradation or delayed collapse.

**(ii) Semantic overlap.** Forgetting is also strongly influenced by semantic overlap among classes in the representation space. As shown in Fig. 3(right), old and new classes often occupy entangled regions rather than well-separated clusters, indicating that their representations are only weakly disentangled. This overlap makes their optimization objectives inherently conflicting during incremental training. Since CISS is dominated by strong gradient signals from newly introduced classes Lopez-Paz & Ranzato (2017); Farajtabar et al. (2020), updates for new classes can easily interfere with old-class representations that no longer receive direct supervision. As a result, old-class features are gradually displaced from their original semantic regions, leading to accumulated representation drift and eventual IoU collapse.

Overall, these findings suggest that NUF is not a random byproduct of incremental training, but a structured phenomenon jointly shaped by class-specific semantic complexity, inter-class feature entanglement, and the

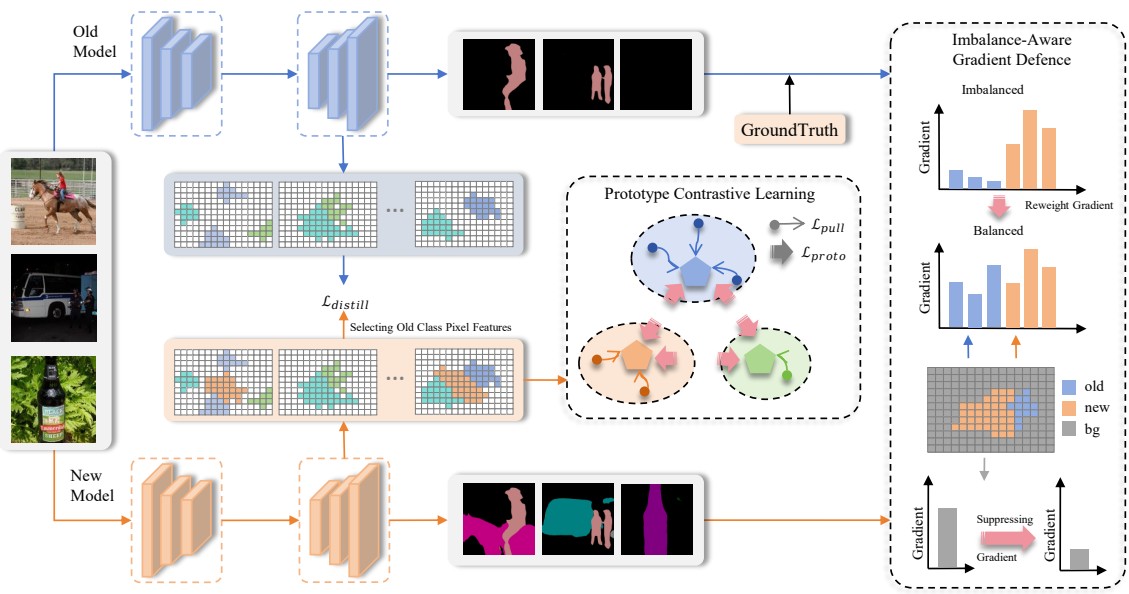

Figure 4: Overview of the proposed framework. It consists of two components: (1) **Imbalance-Aware Gradient Defence (IGD)**, which performs gradient-aware reweighting to strengthen supervision for vulnerable old classes while suppressing the dominance of background and newly introduced classes, thereby mitigating NUF from the optimization perspective; and (2) **Representation Drift Suppressor (RDS)**, which alleviates representation-level forgetting by applying selective decoder-level feature distillation to pseudo-labeled old-class features and prototype-based contrastive learning to improve inter-class separability and stabilize old-class representations.

inherent supervision imbalance of CISS. From a mechanistic perspective, these factors manifest in two tightly coupled forms: biased gradient competition, which causes old-class signals to be progressively diluted during optimization, and representation drift, which destabilizes old semantics across incremental steps. This observation motivates us to address NUF from two complementary perspectives in the proposed method: gradient-level correction to rebalance old–new optimization, and representation-level stabilization to preserve old semantic structure over time.

## 4 Methodology

### 4.1 Problem Formulation

We formulate CISS as a continual learning process over $T$ incremental steps. At each step $t \in \{1, \ldots, T\}$, the model receives a dataset $\mathcal{D}^t = \{(x_i, y_i)\}$, where $x_i \in \mathbb{R}^{H \times W \times C}$ denotes an input image and $y_i \in \mathbb{R}^{H \times W}$ denotes the corresponding segmentation label map. Let $\mathcal{C}^t$ denote the set of new semantic classes introduced at step $t$, and let $\mathcal{C}^{0:t-1} = \bigcup_{k=0}^{t-1} \mathcal{C}^k$ denote the set of all previously learned classes. These class sets are disjoint, i.e., $\mathcal{C}^t \cap \mathcal{C}^{0:t-1} = \emptyset$.

During training at step $t$, only pixels belonging to $\mathcal{C}^t$ are annotated, whereas pixels belonging to $\mathcal{C}^{0:t-1}$ and the true background are all assigned a shared background label 0. Hence, for each pixel $p$ at step $t$, $y_i(p) \in \mathcal{C}^t \cup \{0\}$. Moreover, data from previous steps, i.e., $\mathcal{D}^0, \ldots, \mathcal{D}^{t-1}$, are no longer accessible. The central challenge of CISS is therefore to learn the newly introduced classes $\mathcal{C}^t$ while preserving the knowledge of previously learned classes $\mathcal{C}^{0:t-1}$ under the risk of catastrophic forgetting.

### 4.2 Overview

As discussed before, NUF in CISS is driven by two tightly coupled factors: biased gradient competition caused by severe old–new imbalance, and representation drift accumulated across incremental steps. To address these two issues, we propose a unified framework composed of two complementary modules, as illustrated in Fig. 4.

The first module, **Imbalance-Aware Gradient Defence** (IGD), targets the optimization bias induced by the severe imbalance between old and new classes. It strengthens vulnerable old-class supervision through pixel-wise gradient-aware reweighting and channel-wise balancing, while a background-suppression term further reduces spurious activations of newly introduced classes in old-background regions. The second module, **Representation Drift Suppressor** (RDS), targets the gradual drift of previously learned representations that cannot be fully resolved by gradient correction alone. It improves cross-class separability through prototype-based contrastive learning and stabilizes old semantics through selective decoder-level distillation.

These two modules operate in a complementary manner. IGD mainly rebalances learning signals during optimization and alleviates the dilution of old-class updates, whereas RDS constrains the geometry and temporal consistency of the feature space to preserve old semantic structure across steps. Their combination enables the model to mitigate NUF from both optimization and representation perspectives, leading to more balanced and robust incremental segmentation performance.

### 4.3 Imbalance-Aware Gradient Defence

In CISS, optimization is inherently biased toward newly introduced classes, whereas old classes receive only sparse pseudo labels supervision. Our NUF analysis further indicates that this bias is uneven across semantic regions, with fragile or highly entangled classes suffering more severe degradation. This makes fixed and uniform regularization inadequate for effective forgetting mitigation.

To address this issue, we propose **Imbalance-Aware Gradient Defence (IGD)**, which replaces uniform global compensation with fine-grained pixel-level reweighting. Specifically, we construct a pixel-wise gradient proxy to characterize the local optimization response at step $t$, and use it to dynamically modulate the loss so that informative yet vulnerable pixels receive stronger supervision. For pixel $(i,j)$ at step $t$, let $N_{ij}^t$ denote the logit on the corresponding channel, and let $P^t(x_{ij}^t, \theta^t) = \sigma(N_{ij}^t)$ denote the predicted probability. Using channel-wise bce supervision, the gradient proxy is defined as

$$\Gamma_{ij}^t = \frac{\partial \mathcal{L}_{\text{bce}}\big(N_{ij}^t, \hat{y}_{ij}^t\big)}{\partial N_{ij}^t} = \sigma(N_{ij}^t) - \hat{y}_{ij}^t = P^t(x_{ij}^t, \theta^t) - \hat{y}_{ij}^t, \tag{1}$$

where $\hat{y}_{ij}^t$ denotes the pseudo labels of pixel $(i,j)$. For old-class pixels, we directly use the predictions of the model from the previous step as supervision.

Directly using $|\Gamma_{ij}^t|$ may be unstable due to pseudo labels noise and ambiguous boundaries. We therefore apply logarithmic compression and define the pixel-level gradient weight as:

$$w_{ij}^{\text{grad}} = \log\big(|\Gamma_{ij}^t| + 1\big). \tag{2}$$

Since pixel-level modulation alone cannot eliminate the batch-level imbalance between old and new classes, we further introduce a dynamic channel-wise balancing factor. When the number of recovered old-class pixels is smaller than that of new-class pixels in the current batch, the losses on old-class channels are proportionally up-weighted:

$$w_c^{\text{bal}} = \begin{cases} \dfrac{|\mathcal{M}_{\text{old}}|}{|\mathcal{M}_{\text{new}}|} + 1, & 1 \leq c \leq n_{\text{old}} \ \wedge \ |\mathcal{M}_{\text{old}}| < |\mathcal{M}_{\text{new}}|, \\ 1, & \text{otherwise}, \end{cases} \tag{3}$$

where $\mathcal{M}_{\text{old}}$ and $\mathcal{M}_{\text{new}}$ denote the numbers of old-class and new-class pixels in the current batch. The resulting gradient-aware loss is defined as:

$$\mathcal{L}_{\text{cb}} = \frac{1}{|\Omega|} \sum_{(i,j) \in \Omega} \sum_{c=1}^{C} w_c^{\text{bal}} \cdot w_{ij}^{\text{grad}} \cdot \mathcal{L}_{\text{BCE}} \left( P^t(x_{ij}^t, \theta^t), \hat{y}_{ij}^t \right)_c, \tag{4}$$

where $\Omega$ denotes the set of all image pixels.

In addition, incremental training often causes background drift, i.e., spurious activations of new classes in regions that should remain background. To suppress such false positives, we introduce a background-consistency constraint guided by the old model. Let $\Omega_{\text{bg}}$ denote the set of pixels predicted as background by the old model. We impose low responses on all new-class channels in these regions:

$$\mathcal{L}_{\text{bg}} = \frac{1}{|\Omega_{\text{bg}}|} \sum_{(i,j) \in \Omega_{\text{bg}}} \sum_{c=n_{\text{old}}+1}^{n_{\text{old}}+n_{\text{new}}} \mathcal{L}_{\text{bce}} \left( P^t(x_{ij}^t, \theta^t), 0 \right)_c. \tag{5}$$

Finally, the overall IGD objective is defined as

$$\mathcal{L}_{\text{IGD}} = \mathcal{L}_{\text{cb}} + \mathcal{L}_{\text{bg}}. \tag{6}$$

### 4.4 Representation Drift Suppressor

Although IGD alleviates optimization bias under severe pixel imbalance, it does not explicitly constrain the representation space. As incremental training proceeds, old-class features may still drift and overlap with new-class representations, especially under noisy pseudo-label supervision. To address this issue, we introduce **Representation Drift Suppressor (RDS)**, which enhances representation stability through intra-class compactness, inter-class separation, and temporal feature anchoring.

Specifically, we first encourage intra-class compactness to obtain stable class centers from noisy pixel supervision:

$$\mathcal{L}_{\text{pull}} = \frac{1}{N} \sum_{i=1}^{N} \left\| \mathbf{f}_i^t - \mathbf{p}_i^t \right\|_2^2, \tag{7}$$

where $\mathbf{f}_i^t$ denotes the decoder feature at pixel $i$, and $\mathbf{p}_i^t$ denotes the corresponding class prototype.

While $\mathcal{L}_{\text{pull}}$ reduces intra-class variance by pulling pixel features toward their prototypes, it does not guarantee separability among different classes. Under severe pixel imbalance or when some classes are weakly observed, multiple class clusters may still become entangled or collapse into nearby regions. To explicitly preserve inter-class boundaries, we further introduce a prototype-contrastive term that repels prototypes of different classes:

$$\mathcal{L}_{\text{proto}} = \frac{1}{K} \sum_{k=1}^{K} \frac{1}{K-1} \sum_{\substack{j=1 \\ j \neq k}}^{K} \mathcal{L}_{\text{BCE}} \left( \frac{(\mathbf{p}_k^t)^\top \mathbf{p}_j^t}{\tau}, 0 \right), \tag{8}$$

where $\mathbf{p}_k^t$ denotes the $\ell_2$-normalized prototype of the $k$-th class present in the current mini-batch, and $\tau$ is a fixed temperature. This term complements $\mathcal{L}_{\text{pull}}$ by explicitly separating prototypes of different classes, thereby preserving inter-class boundaries and alleviating feature-space overlap that cannot be resolved by gradient reweighting alone.

However, enforcing prototype separation across all visible class pairs also updates the positions of old-class prototypes and their surrounding features. Since old classes are only weakly supervised at the current step, such updates may accumulate across stages and induce representation drift. To provide a stable temporal anchor, we introduce selective feature distillation that aligns old-class features with those from the previous model:

$$\mathcal{L}_{\text{distill}} = \frac{1}{M} \sum_{i=1}^{M} \left[ 1 - \cos(\mathbf{f}_i^t, \mathbf{f}_i^{t-1}) \right], \tag{9}$$

where $\mathbf{f}_i^t$ and $\mathbf{f}_i^{t-1}$ denote the normalized decoder features from the current and previous models, respectively.

By combining inter-class separation, intra-class compactness, and temporal anchoring, the overall RDS objective is given by

$$\mathcal{L}_{\text{RDS}} = \lambda_1 \mathcal{L}_{\text{proto}} + \lambda_2 \mathcal{L}_{\text{pull}} + \lambda_3 \mathcal{L}_{\text{distill}}, \tag{10}$$

where $\lambda_1$, $\lambda_2$, and $\lambda_3$ are weighting coefficients for the three terms.

### 4.5 Loss Function

The overall training objective is formulated as

$$\mathcal{L}_{\text{total}} = \mathcal{L}_{\text{base}} + \mathcal{L}_{\text{IGD}} + \mathcal{L}_{\text{RDS}}, \tag{11}$$

where $\mathcal{L}_{\text{base}}$ denotes the baseline training loss. In our implementation, it follows the DKD formulation and consists of $\mathcal{L}_{\text{kd}}$, $\mathcal{L}_{\text{dkd}}$, and $\mathcal{L}_{\text{ac}}$.

## 5 Experiments

### 5.1 Experimental Setups

**Datasets and Evaluation Metric.** We evaluate the proposed method on the PASCAL VOC Everingham et al. (2012) and ADE20K Zhou et al. (2019) benchmarks. PASCAL VOC contains 10,582 training images and 1,449 validation images annotated with 20 foreground classes and one background class. ADE20K consists of 20,210 training images and 2,000 validation images spanning 150 semantic categories, including both object and stuff classes. All evaluations are conducted on the official validation splits following the protocols in Cermelli et al. (2020); Baek et al. (2022); Douillard et al. (2021). We adopt mean Intersection-over-Union (mIoU) as the evaluation metric.

**Protocols.** We follow the overlapped CISS setting used in Douillard et al. (2021); Cha et al. (2021), which is generally considered more realistic than the disjoint variant Cermelli et al. (2020); Michieli & Zanuttigh (2021). In the overlapped setting, training images at each incremental step may still contain pixels from previously learned classes or future unseen classes, but only pixels belonging to the currently introduced classes are annotated, while all the remaining pixels are merged into the background label. By contrast, the disjoint setting excludes such overlapping semantics from the current training images, resulting in a cleaner but less realistic data distribution. Each dataset is evaluated under several $X$–$Y$ protocols, where $X$ classes are learned at the base step and $Y$ new classes are introduced at each subsequent step. For example, VOC 19–1 learns 19 base classes and then adds one class per incremental step, whereas VOC 15–1 introduces one new class over six incremental steps.

**Implementation Details.** Our framework is built on DeepLabV3 Chen et al. (2017) with a ResNet-101 He et al. (2016) backbone pretrained on ImageNet Deng et al. (2009). Following Cermelli et al. (2020); Baek et al. (2022), we adopt dataset-specific training schedules. On PASCAL VOC, the model is trained for 60 epochs at each step with a batch size of 24 and polynomial learning rates of 0.001 for the base step and 0.0001 for incremental steps. On ADE20K, we train for 100 epochs with a batch size of 16, using a polynomial learning-rate schedule with linear warm-up Goyal et al. (2017) and initial learning rates of 0.0025 and 0.00025 for the base and incremental steps, respectively. SGD with a momentum of 0.9 is used in all experiments.

### 5.2 Quantitative Evaluation

We evaluate the proposed method on PASCAL VOC 2012 and ADE20K. Tables 1 and 2 report mIoU (%) under the overlapped setting, compared with representative CISS baselines and recent methods, including MiB, PLOP, SSUL, DKD, MicroSeg Zhang et al. (2022b), UCD Yang et al. (2022), GSC, IDEC Zhao et al. (2023), LAG, SVSRD and MLKD Wang et al. (2025).

Table 1: Quantitative mIoU (%) results on PASCAL VOC under overlapped settings. [*] indicates results reproduced by us.

| Method | 19–1(2 steps) | | | 15–5(2 steps) | | | 15–1(6 steps) | | | 10–1(11 steps) | | |
|---|---|---|---|---|---|---|---|---|---|---|---|---|
| | 0–19 | 20 | all | 0–15 | 16–20 | all | 0–15 | 16–20 | all | 0–10 | 11–20 | all |
| MiB[*] | 68.90 | 27.60 | 66.93 | 76.35 | 47.78 | 69.55 | 39.71 | 15.83 | 34.02 | 10.64 | 10.78 | 10.70 |
| PLOP | 75.35 | 37.35 | 73.54 | 75.73 | 51.71 | 70.09 | 65.12 | 21.11 | 54.64 | 44.03 | 15.51 | 30.45 |
| SSUL | 77.73 | 29.68 | 75.44 | 77.82 | 50.10 | 71.22 | 77.31 | 36.59 | 67.61 | 71.31 | 45.98 | 59.25 |
| DKD | 77.76 | 41.52 | 76.03 | 78.83 | 58.21 | 73.92 | 78.14 | 42.65 | 69.69 | 73.10 | 46.51 | 60.44 |
| MicroSeg | 78.80 | 14.00 | 75.70 | 80.40 | 52.80 | 73.80 | 80.10 | 36.80 | 69.80 | 72.60 | 48.70 | 61.20 |
| UCD | 75.90 | 39.50 | 74.00 | 75.00 | 51.80 | 69.20 | 66.30 | 21.60 | 55.10 | 42.30 | 28.30 | 35.30 |
| GSC | 76.90 | 42.70 | 75.30 | 78.30 | 54.20 | 72.60 | 72.10 | 24.40 | 60.80 | 50.60 | 17.30 | 34.70 |
| IDEC | – | – | – | 78.01 | 51.84 | 71.78 | 79.96 | 36.48 | 67.32 | 70.74 | 46.30 | 59.10 |
| LAG | – | – | – | 77.33 | 51.76 | 71.24 | 75.00 | 37.52 | 66.08 | 69.56 | 42.62 | 56.73 |
| SVSRD[*] | 77.36 | 42.41 | 75.70 | 78.42 | 59.09 | 73.81 | 74.32 | 34.87 | 64.93 | 65.16 | 43.38 | 54.79 |
| MLKD | 77.10 | 31.30 | 75.00 | 78.80 | 50.70 | 73.10 | 70.60 | 23.40 | 60.30 | – | – | – |
| Ours | 78.88 | 54.56 | 77.72 | 79.53 | 59.29 | 74.71 | 79.96 | 49.36 | 72.68 | 73.94 | 50.50 | 62.87 |

Table 2: Quantitative mIoU (%) results on ADE20K under overlapped settings.

| Method | 100–50(2 steps) | | | 100–10(6 steps) | | | 50–50(3 steps) | | |
|---|---|---|---|---|---|---|---|---|---|
| | 1–100 | 101–150 | all | 1–100 | 101–150 | all | 1–50 | 51–150 | all |
| MiB | 40.52 | 17.17 | 32.79 | 38.21 | 11.12 | 29.24 | 45.57 | 21.01 | 29.31 |
| PLOP | 41.87 | 14.89 | 32.94 | 40.48 | 13.61 | 31.59 | 48.83 | 20.99 | 30.40 |
| SSUL | 41.28 | 18.02 | 33.58 | 40.20 | 18.75 | 33.10 | 48.38 | 20.15 | 29.56 |
| DKD | 42.41 | 22.86 | 35.95 | 41.56 | 19.51 | 34.26 | 48.84 | 26.28 | 33.90 |
| MicroSeg | 41.52 | 21.64 | 34.94 | 40.17 | 18.84 | 33.11 | 48.55 | 24.84 | 32.85 |
| UCD | 42.12 | 15.84 | 33.31 | 40.80 | 15.23 | 32.29 | 47.12 | 24.12 | 31.79 |
| GSC | 42.40 | 19.20 | 34.80 | 40.80 | 17.60 | 32.60 | 46.20 | 30.20 | 33.20 |
| IDEC | 42.00 | 18.20 | 34.10 | 40.30 | 17.60 | 32.70 | 47.40 | 26.00 | 33.10 |
| LAG | 41.64 | 19.73 | 34.34 | 41.00 | 18.69 | 33.56 | 47.69 | 26.12 | 33.31 |
| SVSRD | 42.65 | 23.23 | 36.22 | 41.72 | 19.55 | 34.38 | 49.11 | 26.74 | 34.29 |
| MLKD | 42.00 | 18.80 | 35.70 | 40.40 | 17.90 | 32.90 | 48.60 | 26.40 | 33.80 |
| Ours | 42.33 | 24.36 | 36.38 | 41.35 | 20.71 | 34.52 | 49.12 | 27.50 | 34.80 |

On PASCAL VOC, our method achieves the best overall performance across all four protocols, and its advantage becomes more evident as the incremental chain grows longer. This trend is particularly meaningful in CISS, where long incremental sequences usually intensify catastrophic forgetting through repeated parameter drift and accumulated pseudo labels errors. In the 15–1 setting, compared with DKD, our method improves the mIoU of old classes, new classes, and all classes by 1.82%, 6.71%, and 2.99%, respectively. These simultaneous gains indicate that the proposed method better balances plasticity and stability, since improvements on new classes in CISS are often accompanied by degraded retention of old ones. In the more challenging 10–1 setting, the corresponding gains further reach 0.84%, 3.99%, and 2.43%, confirming the effectiveness of our method in mitigating forgetting while preserving strong adaptability to newly introduced classes.

A similar trend is observed on ADE20K, where the larger label space and denser semantic overlap make incremental learning substantially more challenging. Compared with DKD, our method improves the mIoU over newly introduced classes and all classes by 1.16% and 0.14%, respectively, in the 100–10 setting, while maintaining comparable performance on previously learned classes. In the 50–50 setting, the corresponding improvements reach 0.76% and 0.90%. These results suggest that the proposed method remains effective even under stronger class interference and greater semantic ambiguity, where the conflict between old-class retention and new-class adaptation is typically more pronounced. Overall, the results on ADE20K further demonstrate that our method can alleviate catastrophic forgetting while maintaining satisfactory plasticity in more challenging CISS scenarios.

The qualitative comparisons in Fig. 5 further support this observation. Compared with previous methods, our approach yields more stable predictions on previously learned classes, with fewer fragmented regions and

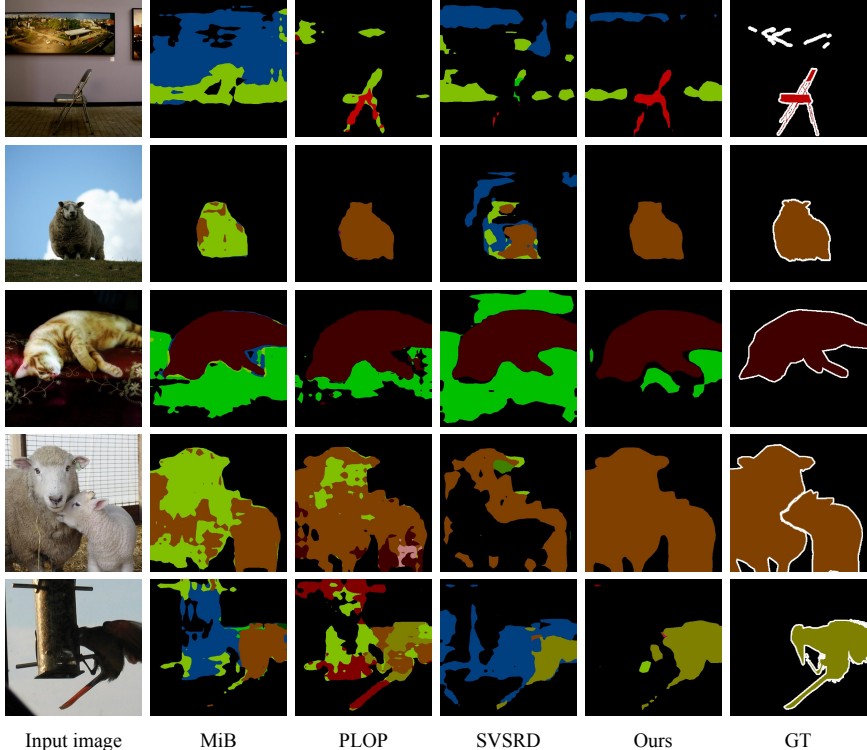

|  Input image | MiB | PLOP | SVSRD | Ours | GT |

Figure 5: Qualitative comparison of our method and prior approaches on Pascal VOC 2012 under the 15-1 overlapped setting.

fewer misclassifications caused by interference from newly introduced classes. This suggests that the proposed method more effectively suppresses catastrophic forgetting while maintaining sufficient discriminability for new-class adaptation.

### 5.3 Ablation Studies

Table 3 presents the component ablation on VOC 15–1 (6 steps), where IGD and RDS are evaluated individually and jointly using DKD as the baseline. RDS alone yields only modest gains, with the best variant improving the overall mIoU by 1.29% and the mIoU over newly introduced classes by 1.57%. Moreover, using the pull term alone causes a clear performance drop, indicating that enforcing feature compactness without sufficient class discrimination can instead damage representation structure in CISS. Although adding prototype contrast and distillation gradually restores performance, the overall improvement remains limited, suggesting that representation regularization alone is insufficient to effectively counter long-chain forgetting.

In contrast, IGD provides the dominant source of improvement, increasing the overall mIoU by 2.04% and the mIoU over newly introduced classes by 3.40%. This result highlights that correcting gradient imbalance is more fundamental in CISS, where optimization is easily dominated by newly introduced classes and thus prone to aggravating forgetting of old ones. After further introducing background suppression, performance improves again, indicating that better control of old–new competition leads to a more favorable stability–plasticity trade-off.

More importantly, RDS becomes clearly more effective when combined with IGD. Built on the strongest IGD-only variant, the full model further improves the mIoU over newly introduced classes by 1.02% and the overall mIoU by 0.27%, while preserving essentially unchanged performance on previously learned classes. This observation confirms that the two modules act in a complementary manner: IGD stabilizes optimization

Table 3: Component ablation studies on VOC 15-1 (6 steps).

| Loss components | | | | | | mIoU (%) | | |
|---|---|---|---|---|---|---|---|---|
| $\mathcal{L}_{mbce}$ | $\mathcal{L}_{cb}$ | $\mathcal{L}_{bg}$ | $\mathcal{L}_{proto}$ | $\mathcal{L}_{pull}$ | $\mathcal{L}_{distill}$ | 0-15 | 16-20 | all |
| ✓ | | | | | | 78.18 | 42.57 | 69.70 |
| ✓ | | | ✓ | | | 78.77 | 43.37 | 70.34 |
| ✓ | | | | ✓ | | 76.42 | 38.19 | 67.32 |
| ✓ | | | | | ✓ | 79.32 | 41.84 | 70.40 |
| ✓ | | | ✓ | ✓ | | 79.00 | 43.46 | 70.54 |
| ✓ | | | ✓ | ✓ | ✓ | 79.80 | 44.14 | 70.99 |
| | ✓ | | | | | 79.80 | 45.97 | 71.74 |
| | ✓ | ✓ | | | | 79.93 | 48.34 | 72.41 |
| | ✓ | | ✓ | | | 79.11 | 48.18 | 71.74 |
| | ✓ | | | ✓ | | 77.72 | 41.97 | 69.21 |
| | ✓ | | | | ✓ | 79.88 | 45.97 | 71.81 |
| | ✓ | | ✓ | ✓ | | 79.60 | 48.22 | 72.13 |
| | ✓ | | ✓ | ✓ | ✓ | 79.82 | 47.99 | 72.24 |
| | ✓ | ✓ | ✓ | ✓ | ✓ | **79.96** | **49.36** | **72.68** |

and mitigates forgetting accumulation, whereas RDS further enhances representation consistency once the training process has been properly conditioned.

### 5.4 Parameter Sensitivity

We analyze the sensitivity of the three RDS weights, where $\lambda_1$, $\lambda_2$, and $\lambda_3$ correspond to $\mathcal{L}_{\text{proto}}$, $\mathcal{L}_{\text{pull}}$, and $\mathcal{L}_{\text{distill}}$, respectively, as reported in Table 4. Among them, $\lambda_1$ is the most influential, since it directly controls inter-class separation. A small $\lambda_1$ is insufficient to resolve semantic overlap, whereas an overly large one causes the contrastive term to dominate optimization and disrupt the balance with compactness and distillation. By comparison, the model is less sensitive to $\lambda_2$, indicating that moderate intra-class compactness is sufficient, while excessive compression may instead weaken discriminability. The effect of $\lambda_3$ reflects the trade-off between preserving old representations and maintaining flexibility for new-class learning: increasing $\lambda_3$ helps suppress representation drift, but an overly strong distillation constraint may impair plasticity. These results suggest that RDS is effective only when class separation, intra-class compactness, and old-feature preservation are properly balanced. Accordingly, we set $\lambda_1 = 0.1$, $\lambda_2 = 1$, and $\lambda_3 = 5$ on VOC. Following the same principle, we adopt a larger $\lambda_1$ on ADE20K to account for stronger semantic overlap, and further increase $\lambda_3$ in long-step settings to better counter accumulated representation drift.

Table 4: Parameter sensitivity of $\lambda_1$, $\lambda_2$, and $\lambda_3$ on VOC 15–1 (6 steps).

| $\lambda_1$ | $\lambda_2$ | $\lambda_3 = 1$ | $\lambda_3 = 3$ | $\lambda_3 = 5$ | $\lambda_3 = 10$ |
|---|---|---|---|---|---|
| | 1 | 72.55 | 72.60 | **72.68** | 72.65 |
| 0.1 | 3 | 70.89 | 71.00 | 71.18 | 71.21 |
| | 5 | 67.21 | 68.05 | 68.34 | 68.67 |
| | 1 | 72.20 | 72.26 | 72.23 | 72.23 |
| 0.2 | 3 | 71.03 | 71.21 | 71.26 | 71.20 |
| | 5 | 68.84 | 69.11 | 69.05 | 69.31 |
| | 1 | 63.33 | 64.19 | 66.77 | 70.59 |
| 1.0 | 3 | 61.50 | 65.42 | 68.45 | 70.25 |
| | 5 | 59.51 | 65.53 | 68.13 | 68.98 |

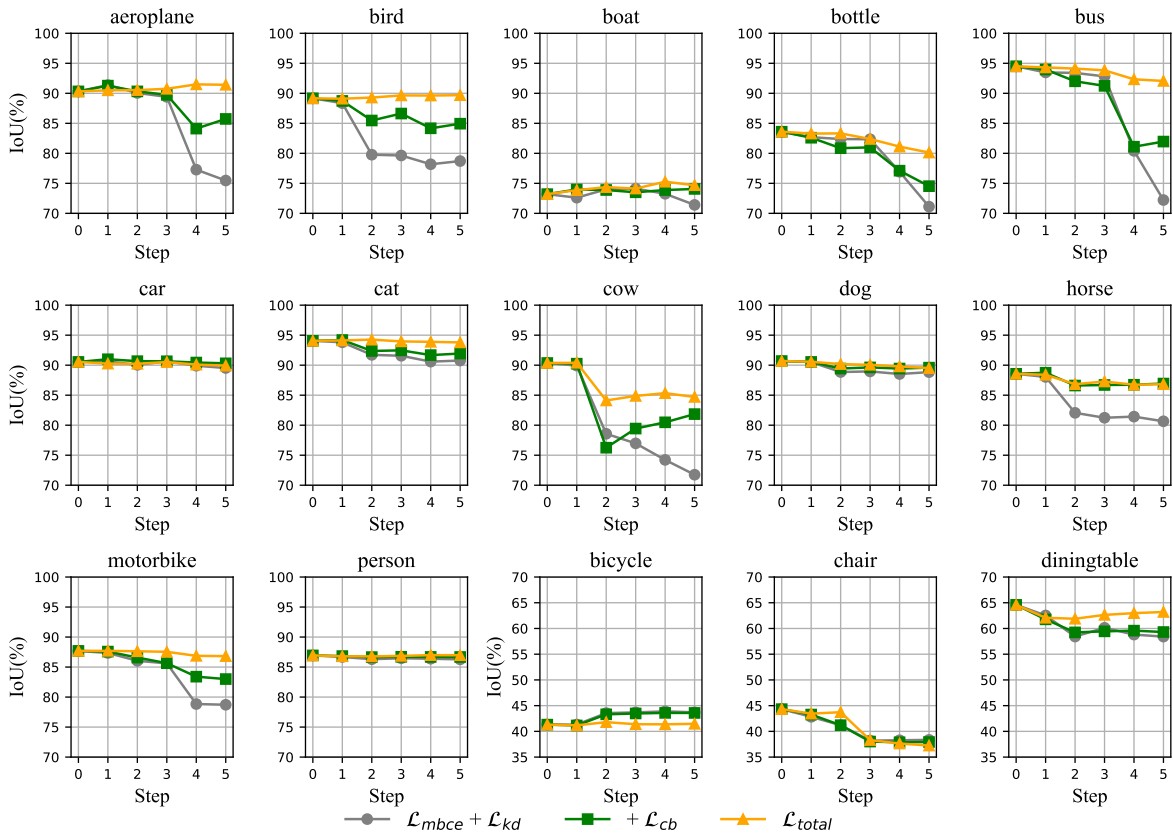

Figure 6: Visualization of the mIoU progression across incremental steps on VOC 15-1 (6 steps).

## 5.5 Discussion

To more intuitively examine forgetting on previously learned classes, Fig. 6 shows the class-wise IoU trajectories of all base classes over incremental steps under the VOC 15–1 protocol. The gray, green, and orange curves correspond to the baseline $\mathcal{L}_{\mathrm{mbce}} + \mathcal{L}_{\mathrm{KD}}$, the variant augmented with $\mathcal{L}_{\mathrm{cb}}$, and the full model $\mathcal{L}_{\mathrm{total}}$, respectively. The trajectories clearly reveal non-uniform forgetting: while some base classes remain relatively stable, vulnerable classes such as *bus*, *cow*, *aeroplane*, and *motorbike* undergo abrupt collapse or persistent degradation under the baseline. After introducing $\mathcal{L}_{\mathrm{cb}}$, these classes generally exhibit slower decline or clearer recovery, indicating that gradient-aware correction can provide targeted compensation for high-risk old classes.

Notably, this compensation often appears with a delay. In the early stage after a class becomes old, its pixels usually occur only as sparse co-occurring regions in the current training data, while optimization is dominated by abundant and strongly supervised pixels from newly introduced classes. As a result, although reweighting amplifies the gradients of vulnerable old-class pixels, the net compensation remains limited at first and is reflected mainly in slowing further deterioration rather than in immediate recovery. As incremental learning proceeds, the interfering new classes from earlier steps also become old classes, and the relative occurrence frequencies of the competing semantics become more comparable. Under this condition, the reweighting mechanism can operate more effectively, so the compensation effect gradually accumulates and becomes visible as recovery in later steps.

The full model further improves over the $\mathcal{L}_{\mathrm{cb}}$-only variant on most classes and yields smoother and more persistent retention trends, indicating that the additional stabilization constraints complement gradient-aware correction by providing earlier protection before the compensation effect is fully accumulated. Overall,

Fig. 6 verifies that the proposed method improves old-class preservation and that the two components provide complementary gains in both short- and long-horizon incremental learning.

## 6 Conclusion

In this work, we investigate non-uniform forgetting in class-incremental semantic segmentation and show that previously learned classes can exhibit markedly different degradation magnitudes, collapse timing, and temporal dynamics during incremental training. To address this issue, we propose a pseudo-label-assisted framework with two complementary modules. IGD alleviates the optimization bias caused by severe old–new pixel imbalance through gradient-aware reweighting and background suppression, whereas RDS stabilizes old semantics by improving feature-space separability and suppressing cross-step representation drift via prototype-based contrastive learning and selective decoder-level distillation. Extensive experiments on multiple benchmarks and incremental protocols demonstrate that the proposed method consistently improves old-class retention and overall incremental balance. These results validate the effectiveness of our framework and highlight the importance of explicitly addressing heterogeneous forgetting in continual semantic segmentation.

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
