# OpenReview forum: "Mitigating Non-Uniform Forgetting Dynamics for Class-incremental Semantic Segmentation"
_TMLR — Under review for TMLR_

### Review · Reviewer_JDDW · 2026-06-18

**Summary Of Contributions:**

The manuscript begins by showing empirically that, in class-incremental semantic segmentation (CISS), previously learned classes are forgotten in a non-uniform manner. The authors then conduct a study on the reasons behind this non-uniformity and use their findings to propose a method for CISS that addresses the root causes they identify. Their approach trains using a loss function with multiple components: pixel-wise up-weighting of losses based on their gradients, class balancing, background suppression, contrastive learning of class representations, and temporal stability. Empirical results demonstrate the strengths of the proposed method, accompanied by ablation studies that inspect the relative merits of its various components.

######## Strengths ########

- The manuscript identifies the problem of non-uniform forgetting specifically in the context of CISS. The motivation for this, validated empirically, is exceptionally well conveyed in the introduction of the manuscript.
- The experimental setting is well designed and the proposed method exhibits very strong results across a variety of class-sequence settings.
    - The ablation studies are meaningful and demonstrate that each component of the approach is necessary to achieve peak performance
    - The learning trajectories in Fig. 6 show that the method significantly reduces the non-uniformity of forgetting, validating the authors' design choices.


######## Weaknesses ########

The main shortcoming of the submission is the weak connection between the problem of non-uniform forgetting and the proposed approach to CISS. I see two key issues:
- The evidence backing the semantic complexity/overlap arguments is not strong.
    - The definition of semantic complexity is too loose. The argument that "aeroplanes" have higher diversity than "persons" is unsubstantiated and would require some compelling metric to justify.
    - What 2D representation is used to visualize semantic overlap? How are the representations learned?
    - Fig 3 doesn't quite show whether the classes that overlap map to those that exhibit higher degrees of forgetting.
    - There are many alternative hypotheses that could explain the non-uniformity in forgetting: classes that are more/less represented in their original training sets, classes that are more/less represented (either annotated or not) in the new training sets, classes that are learned more/less easily (e.g., see [1]). The evidence in Sec 3.2 is insufficient to support the claim that semantic complexity and overlap explain non-uniformity.
    - "NUF is ... a structured phenomenon ... shaped by ... semantic complexity, inter-class feature entanglement, and the inherent supervision imbalance of CISS... [T]hese factors manifest [as] biased gradient competition ... and representation drift." — It is completely unclear to me how this paragraph relates to the remainder of this section. When was "supervision imbalance" discussed and demonstrated to have an effect on the non-uniformity of forgetting? How do complexity, entanglement, and imbalance result in biased gradient competition? How do they result in representation drift? The arguments here are insufficient.
- The technical details of the approach and their rational are horribly unclear
    - Sec 4.3
        - The explanation throughout this section is unclear and no explicit rationale is provided for the choices made throughout.
        - "let $N^{t}_{ij}$ denote the logit on the corresponding channel" — Which corresponding channel?
        - Does $\sigma({\cdot})$ denote the sigmoid? I guess it does because the authors also discuss binary cross-entropy. Is the assumption that one pixel may correspond to multiple semantic masks?
        - If $\mathcal{L}\_\mathrm{bce} = \mathcal{L}\_\mathrm{BCE}$, which is not clear because these are never defined and they are written in two slightly different notations, then why apply a weight based on the gradient of the same loss computed in Eq. 4?
        - Overall, the authors do not motivate the use of a gradient-based per-pixel weight: what does it achieve on the objective? How does it achieve that?
        - How does $w_{c}^\mathrm{bal}$ achieve balancing that differentiates across old classes non-uniformly affected by forgetting? It seems to just assign the same weight to all old classes.
        - I also don't understand why an additional explicit background loss is needed. Doesn't the original cross-entropy loss include precisely the terms in Eq. 5, since the background pixels have negative labels for the foreground classes?
    - Sec 4.4
        - The authors have repeatedly used the term "pixel imbalance". What does that mean? "Class imbalance" refers to certain classes being over/under-represented in the dataset. Does "pixel imbalance" somehow refer to certain pixels being over/under-represented in the dataset? I don't even know what that would mean, let alone how "IGD alleviates optimization bias under severe pixel imbalance"
        - "old-class features may still drift and overlap with new-class representations" — Could they also overlap with other old-class representations, or background representations?
        - The class prototypes in Eq 7 are undefined. Are they obtained during initial training and kept fixed, or updated incrementally as features change over time?
        - If we need "temporal anchoring" from Eq. 9, wouldn't it be reasonable to use fixed prototypes instead? This may be advantageous computationally, since Eq. 9 requires two forward passes, whereas maintaining fixed prototypes requires only storing one prototype vector per class. A discussion on this that supports the choice to separate these two issues would be most welcome.
    - Sec 4.5
        - The three loss terms that make up $\mathcal{L}_\mathrm{base}$ are undefined.

In addition, here are some additional concerns that I believe would need to be addressed before publication:
- I'm not quite sure that peak + variance are the correct metrics to capture temporal patterns—e.g., the same peak and variance would arise from mirrored signals. Also, the fact that the trend is linear shows that these two quantities do not (in this example) provide additional information.
- The analyses in text for experimental results don't always match the quantitative results:
    - "its advantage becomes more evident as the incremental chain grows longer" — Can the authors clarify exactly how readers should interpret this statement? As far as I can tell, the gains in "all" w.r.t. the second-best competing methods are fairly stable across 2-step, 6-step, and 11-step sequences (in Table 1), and 2-step, 3-step, and 6-step sequences in Table 2.
    - The analysis about $\lambda_1$ seems to be unsupported by Table 4: the highest performance values are with the smallest $\lambda_1$ values reported.
    - The comparison to $\lambda_2$ appears to be similarly unsupported: the drops caused by $\lambda_2$ are similar to those caused by $\lambda_1$, and the larger drops seen for $\lambda_1=1.0$ compared to $\lambda_2=5$ may be due to the fact that the former was increased by a factor of 10 and the latter by a factor of 5.
- "Accordingly, we set..." — Does this entail that values for $\lambda$ are obtained via a grid-search on final performance? It is well known that this violates continual learning assumptions, as it is necessary for the training process to see future classes before actually beginning the training.

[1] Hacohen & Tuytelaars. "Predicting the Susceptibility of Examples to Catastrophic Forgetting", 2024.

**Additional Comments:**

The following points are provided as feedback to hopefully help better shape the submitted manuscript, but will not impact my recommendation in a major way.

Abstract
- My first question immediately is: do the reasons that cause NUF in CISS not affect CIL for, say, classification?

Intro
- Upon reading the first few paragraphs, I wondered: is the assumption that old classes are annotated as background standard in the CISS literature? I'm wondering what the real-world setting for this would be. Of course, annotating all new images with each old class seems pretty expensive. But it's interesting because it is unlike other CIL settings where there are no *inputs* corresponding to past classes.
    - This is, much later, somewhat addressed by referencing prior work. A comment in the introduction about how this setting was identified in prior literature and matches some realistic scenario would strengthen the introduction.
- Fig 1: for an illustrative experiment this level of detail is fine, but I hope to see details of the setup somewhere later in the paper. The introduction should also refer the reader to those details. I'm wondering: Is training just naive finetuning? Do the images for subsequent steps include the categories that are more/less forgotten?
- Given the mismatch in the CIL vs CISS problem settings, I would be very interested in a result that demonstrated that these dynamics are different for CIL. In particular, I suspect that old classes that show up in new images annotated as background make forgetting significantly worse than in CIL.
- The problem setting and motivation for it (paragraphs 1–3) are excellent.
- The *method* description is very short and full of jargon: "old-new imbalance", "pixel-wise gradient-aware reweighting", "channel-wise balancing", "background-suppression term", "spurious activations", "prototype-based contrastive learning", "selective decoder-level distillation". It would be far more helpful to provide a more intuitive description of the key elements of the solution, and how they address the challenges brought on by NUF or other (perhaps secondary) phenomena.

Sec 2
- Key missing references: [2, 3]
- The citation format throughout this section is quite bad: no parentheses (\citep) for parentheticals, repeated author names for non-parentheticals (\citet), this inexplicable error "Yin et al. Xu et al. (2024)", "He (2024)" is a single author so no "et al." should follow and it should be singular... [This issue persists throughout the manuscript.]

Sec 3.1
- $L_\mathrm{mbce}$ and $L_\mathrm{KD}$ have not been defined. An English description is needed to convey the "baseline".

Sec 4.1
- Would it not be more accurate to define the label set as either $\mathbb{N}^{H \times W}$ or ${\mathcal{C}^{t}}^{H \times W}$?

Sec 4.2
- The second paragraph here repeats, nearly verbatim, what has been said at least 3 times so far. While repetition of concepts in different contexts is very often a useful writing tactic, repeating the exact same words again and again, without adding any additional context, is rarely useful.

Sec 5.3
- Ablations are nice

Typos/style/grammar/notation
- Sec 4.3 — "bce supervision" -> "binary cross-entropy supervision"
- Is $L\_\mathrm{MBCE} = \mathcal{L}\_\mathrm{bce} = \mathcal{L}\_\mathrm{BCE}$?

[2] Ramasesh et al. "Anatomy of catastrophic forgetting: Hidden representations and task semantics", 2021.

[3] Lee et al. "Continual learning in the teacher-student setup: Impact of task similarity", 2021.

**Audience:**

Yes

**Audience Explanation:**

There is a significant continual learning community, and CISS seems to fit well within that theme.

**Claims And Evidence:**

No

**Claims Explanation:**

My main concerns regarding claims are:
- The weakness of the evidence supporting the claim that semantic complexity and semantic overlap are causes for non-uniform forgetting
- The lack of rationale behind the design choices of the proposed algorithm
- The apparent disconnect between quantitative results and their analyses
- The choice of hyperparameters based on best performance across the sequence

**Requested Changes:**

See "Weaknesses".

---

### Review · Reviewer_2kyj · 2026-06-18

**Summary Of Contributions:**

The paper addresses class-incremental semantic segmentation, where a model must learn newly introduced classes while retaining performance on previously learned ones. Its main observation is that forgetting is not uniform across old classes: some classes remain relatively stable, while others degrade much earlier or more severely. To address this, the authors propose a pseudo-label-assisted framework with two components: IGD, which rebalances optimization through gradient-aware reweighting and old/new class balancing, and RDS, which stabilizes representations using prototype-based contrastive learning and feature distillation. The results show consistent improvements over several prior CISS methods, with the strongest gains appearing on PASCAL VOC. The ablation study also supports the contribution of the proposed components, especially IGD. Overall, the paper presents a clear and practically motivated approach, with stronger empirical support for improved CISS performance than for fully explaining the causal sources of non-uniform forgetting.

**Audience:**

Yes

**Audience Explanation:**

Yes. The paper studies a relevant problem in class-incremental semantic segmentation and highlights the useful observation that forgetting can vary substantially across old classes.

**Broader Impact Concerns:**

I do not see broader impact concerns.

**Claims And Evidence:**

No

**Claims Explanation:**

- The submission provides reasonable evidence that non-uniform forgetting occurs in CISS, especially through class-wise IoU trajectories and forgetting metrics. This part of the claim is clear and mostly well supported.

- The performance evidence is more mixed. The gains on PASCAL VOC are meaningful, but the improvements on ADE20K are relatively small, which makes the overall strength of the empirical support less convincing.

- The claimed causes of non-uniform forgetting, such as semantic complexity and semantic overlap, are not fully established. The supporting evidence is mostly qualitative, and alternative factors such as class frequency, co-occurrence patterns, pseudo-label quality, and intra-class diversity are not sufficiently analyzed.

- Overall, the results suggest that the proposed method is promising, but the evidence does not yet fully support the broader mechanistic claims behind the approach.

**Requested Changes:**

- I would encourage the authors to strengthen the analysis of what drives non-uniform forgetting. For example, in PASCAL VOC, the `person` class may behave differently because it is the only human class and may also co-occur frequently with classes such as `bicycle`, `motorbike`, or `chair`. A class-level analysis of frequency, co-occurrence, and semantic proximity could help quantify whether the observed forgetting patterns are due to semantic complexity/overlap or dataset-specific class structure.

- Relatedly, the causal explanation behind NUF should be made more convincing. The current discussion around semantic complexity and semantic overlap is intuitive, but mostly qualitative. Quantitative measures such as class frequency, intra-class feature variance, prototype distance, feature overlap, pseudo-label quality, or gradient similarity would make the proposed explanation much stronger.

- The ADE20K results would benefit from reporting variance across multiple runs. Some of the improvements in overall mIoU are quite small, so standard deviations or confidence intervals would help clarify whether these gains are robust.

- I would also like to see whether the method generalizes beyond the current DeepLabV3/ResNet-101 setup. An additional experiment with a more recent transformer-based segmentation backbone would strengthen the evidence that the method is not tied mainly to a standard CNN-based CISS setting.

Addressing these points would substantially improve my confidence in the claimed mechanism and the generality of the empirical findings, and I would be willing to reconsider the claims/evidence assessment if these analyses are added.

---

### Review · Reviewer_cvvv · 2026-06-19

**Summary Of Contributions:**

The paper studies class-incremental semantic segmentation (CISS) and makes two claims: an observation that old classes forget non-uniformly (NUF), differing in severity, onset, and pattern, quantified by three per-class statistics $F_c$, $P_c$, $V_c$ (Sec. 3) and attributed to semantic complexity, semantic overlap, and old/new imbalance; and a method to mitigate it, combining IGD (gradient-magnitude reweighting, old/new channel balancing, background suppression) and RDS (prototype pull, prototype repulsion, cosine distillation). It reports modest gains over recent CISS baselines on PASCAL VOC and ADE20K (Tables 1-2), with ablations and a sensitivity study.

**Audience:**

Yes

**Audience Explanation:**

The NUF framing and the per-class temporal forgetting metrics ($F_c$, $P_c$, $V_c$) are of clear interest to the CISS and continual-learning community, independent of novelty or SOTA. The observation that mIoU smooths over class-wise forgetting dynamics is a useful point on its own.

**Broader Impact Concerns:**

None. This is a methods paper on standard public benchmarks (PASCAL VOC, ADE20K); no Broader Impact Statement is needed.

**Claims And Evidence:**

No

**Claims Explanation:**

The narrow factual claims hold (forgetting is non-uniform; VOC mIoU improves). But the two contributions the paper advertises are not adequately supported.

**1. The method is not shown to reduce non-uniformity specifically.**

- The paper defines $F_c$, $P_c$, $V_c$ in Sec. 3 but never applies them to the method: every result reports the old/new/all mIoU aggregates (Tables 1-3) the authors describe as concealing heterogeneity, with no post-method dispersion number. Fig. 6 shows a broadly uniform lift across classes, which raises the average but does not indicate the spread narrowed.
- No term in IGD or RDS is class- or vulnerability-aware: Eq. 2-9 depend on per-pixel error or treat all old classes alike, never on $F_c$/$P_c$, so the same method would apply under uniform forgetting.
- The old-class improvement is smaller than the new-class improvement (VOC 15-1: 1.82 old vs 6.71 new; 10-1: 0.84 vs 3.99), so most of the reported gain is on new classes rather than on old-class retention. The ablation is consistent with this: $L_{pull}$ alone lowers old-class mIoU (78.18→76.42, Table 3), and full vs best IGD-only leaves old-class mIoU essentially unchanged.

**2. The causal analysis is asserted rather than established.** That classes forget at different rates is largely expected from imbalance and difficulty, and heterogeneous/imbalanced forgetting is already studied in the classification work the paper cites (He 2024; Dong et al. 2023; Xu et al. 2024). The explanation in Sec. 3.2 (semantic complexity and overlap) rests on one illustrative figure (Fig. 3), with no quantitative link between a measure of complexity or overlap and the per-class forgetting statistics.

These are the issues that most affect my assessment. Smaller evidence gaps (no seeds/variance; a possible error in Eq. 3; undefined prototypes) are listed under Requested Changes.

**Requested Changes:**

**Critical** (needed to support the central claims):

1. Show the method reduces non-uniformity using the paper's own metrics: report $F_c$/$P_c$/$V_c$ (or $\mathrm{std}(F_c)$ and the vulnerable-vs-stable gap) for baseline vs. full method, and show vulnerable classes recover more than already-stable ones.
2. Either add a mechanism keyed to per-class vulnerability and show it outperforms the class-agnostic version, or reframe the paper as a generic CISS method analyzed through an NUF lens, resting the contribution on that.

**Would strengthen the work:**

3. Substantiate the Sec. 3.2 causal story by quantifying the relationship between semantic complexity / overlap and the per-class forgetting statistics, rather than illustrating it with one figure.
4. Add seeds and variance for the main tables, ADE20K especially, where the "all" gains over SVSRD (0.16, 0.14, 0.51; Table 2) are hard to separate from run-to-run noise.
5. Clarify or fix Eq. 3. As written, the old-channel balance weight is (number of old pixels / number of new pixels) + 1, which tends to 1 as old pixels become scarce, so the correction weakens precisely when the old/new imbalance is most severe. An inverse-frequency form (number of new pixels / number of old pixels) would match the stated intent. Report its effect on the IGD ablation.
6. Define the prototypes in Eq. 7-8 (running mean, per-batch, learned?), and justify the BCE-to-0 all-pairs repulsion against a standard contrastive loss, noting its behavior as the class count grows (all-pairs separation is not satisfiable for ADE20K's 150 classes).
7. Tone down "first to characterize NUF" relative to the cited classification continual-learning literature on heterogeneous/imbalanced forgetting.